# Prediction of Prostate Cancer Disease Aggressiveness Using Bi-Parametric Mri Radiomics

**DOI:** 10.3390/cancers13236065

**Published:** 2021-12-01

**Authors:** Ana Rodrigues, João Santinha, Bernardo Galvão, Celso Matos, Francisco M. Couto, Nickolas Papanikolaou

**Affiliations:** 1Champalimaud Foundation—Centre for the Unknown, 1400-038 Lisbon, Portugal; joao.santinha@research.fchampalimaud.org (J.S.); celso.matos@research.fchampalimaud.org (C.M.); nickolas.papanikolaou@research.fchampalimaud.org (N.P.); 2Faculty of Sciences, University of Lisbon, 1649-004 Lisboa, Portugal; 3Instituto Superior Técnico, University of Lisbon, 1649-004 Lisboa, Portugal; 4Faculty of Sciences and Technology, NOVA University, 2825-149 Caparica, Portugal; bernardo.galvao@novaims.unl.pt; 5LASIGE, Faculty of Sciences, University of Lisbon, 1649-004 Lisboa, Portugal; fjcouto@edu.ulisboa.pt

**Keywords:** radiomics, prostate cancer, machine learning, bi-parametric MRI

## Abstract

**Simple Summary:**

The use of radiomics has been studied to predict Gleason Score from bi-parametric prostate MRI examinations. However, different combinations of type of input data (whole prostate gland/lesion features), sampling strategy, feature selection method and machine learning algorithm can be used. The impact of such choices was investigated and it was found that features extracted from the whole prostate gland were more stable to segmentation differences and produced better models (higher performance and less overfitting). This result suggests that the areas surrounding the tumour lesions offer relevant information regarding the Gleason Score that is ultimately attributed to that lesion.

**Abstract:**

Prostate cancer is one of the most prevalent cancers in the male population. Its diagnosis and classification rely on unspecific measures such as PSA levels and DRE, followed by biopsy, where an aggressiveness level is assigned in the form of Gleason Score. Efforts have been made in the past to use radiomics coupled with machine learning to predict prostate cancer aggressiveness from clinical images, showing promising results. Thus, the main goal of this work was to develop supervised machine learning models exploiting radiomic features extracted from bpMRI examinations, to predict biological aggressiveness; 288 classifiers were developed, corresponding to different combinations of pipeline aspects, namely, type of input data, sampling strategy, feature selection method and machine learning algorithm. On a cohort of 281 lesions from 183 patients, it was found that (1) radiomic features extracted from the lesion volume of interest were less stable to segmentation than the equivalent extraction from the whole gland volume of interest; and (2) radiomic features extracted from the whole gland volume of interest produced higher performance and less overfitted classifiers than radiomic features extracted from the lesions volumes of interest. This result suggests that the areas surrounding the tumour lesions offer relevant information regarding the Gleason Score that is ultimately attributed to that lesion.

## 1. Introduction

Prostate cancer is the second most prevalent cancer in the world, according to the World Health Organization; 1,414,259 new cases were reported in 2020, with an ASR (world) mortality rate of 7.7 per 100,000 [1].

Prostate cancer in its early stages does not cause any specific symptoms, so a suspicion of PCa can arise from: an abnormality on digital rectal examination, DRE [2,3,4], or an elevated level of prostate-specific antigen (PSA) in the serum [3,5]. Both situations are not specific to PCa, making an appearance in conditions like BPH (benign prostatic hypertrophy or enlargement of the prostate) and prostatitis (inflammation of the prostate) [6]. Prostate cancer diagnosis and classification is not ideal, relying on these unspecific measures, followed by TRUS-guided biopsy, where an aggressiveness level is attributed in the form of Gleason Score (GS).

The Gleason Score is the most widely used measure for PCa aggressiveness. This grading system is assigned to a lesion after biopsy, according to the microscopic appearance of the tissue. Two grades, ranging from 1 to 5, are given per patient. The primary grade represents the GS of the largest area of the tumour and the secondary grade describes the GS of the next largest area. The sum of the two scores is taken to be the final GS. The larger the GS the more likely it is that the cancer will grow and spread quickly, with a high GS lesion being considered clinically significant [7].

Contrary to TRUS, magnetic resonance images (MRI) allow for a clear visualization of the zonal anatomy of the prostate [8]. Multiparametric MRI (mpMRI) is a combination of functional and anatomical imaging methods: T1-weighted imaging (T1W), T2-weighted imaging (T2W), diffusion weighted imaging (DWI) and dynamic contrast enhanced MRI (DCE-MRI) [8]. mpMRI is able to provide morphologic and metabolic data as well as characterize tissue vascularity, showing promise in the detection of PCa [9,10]. In this study, bi-parametric MRI was utilized, which comprises of T2W and DW sequences. The use of bpMRI has clear advantages in the reduction of image acquisition time and overall costs per patient [11].

One of the biggest challenges in the clinical use of mpMRI, and consequently bpMRI, is that its interpretation is dependent on the radiologist’s subjective opinion and, thus, is inevitably affected by a high rate of inter-reader variability [12]. In order to reduce these effects, a standardized reporting system was developed, the Prostate Imaging Reporting and Data System (PI-RADS) [13], which assigns a specific score of suspicion to the MRI sequences. Despite this, there is still room for improvement in mpMRI reporting. Hence, efforts have been made to implement computer-aided diagnosis (CAD) coupled with radiomics and machine learning to predict GS from clinical images, with the aim to bypass interobserver variability, showing promising results [14,15,16,17].

Radiomics is the transformation of medical images into high dimension mineable data through the extraction of quantitative features [18]. Based on the hypothesis that tumour tissue characteristics can be quantified by the extracted features, these can be used to build supervised machine learning models capable of assessing the GS attributed to the referred image. The use of radiomic features extracted from bpMRI can be easily found in the literature, with a meta-analysis by Cuocolo et al. reporting an average AUC of 0.90 [19].

That being said, the main goal of this work was to develop supervised machine learning models exploiting radiomic features extracted from bpMRI examinations (T2W, DWI, and ADC), to predict biological aggressiveness in the form of Gleason Score. In this work, we will address a supervised binary classification machine learning problem, where the input is a vector of radiomic features and the output is the clinical significance of the tumour, described as True for clinically significant PCa or False for clinically non-significant PCa.

## 2. Materials and Methods

### 2.1. Data Description

Our dataset consisted of T2W, DW, and ADC data from the SPIE-AAPM-NCI PROSTATEx challenge (the data can be downloaded from (https://wiki.cancerimagingarchive.net/pages/viewpage.action?pageId=23691656 (accessed on 28 September 2020)) [20,21,22]. The MRI exams were acquired at the Prostate MR Reference Center—Radboud University Medical Centre (Radboudumc) in the Netherlands. Due to the public nature of the data, ethics committee approval was waived for this study. The following description of the dataset was provided by the Challenge’s organizers: “The images were acquired on two different types of Siemens 3T MR scanners, the MAGNETOM Trio, and Skyra. T2-weighted images were acquired using a turbo spin echo sequence and had a resolution of around 0.5 mm in plane and a slice thickness of 3.6 mm. The DWI series were acquired with a single-shot echo planar imaging sequence with a resolution of 2-mm in-plane and 3.6-mm slice thickness and with diffusion-encoding gradients in three directions. Three b-values were acquired (50, 400, and 800), and subsequently, the apparent diffusion coefficient (ADC) map was calculated by the scanner software. All images were acquired without an endorectal coil”. The dataset is composed of 281 lesions from 183 patients. The approximate location of the centroid of each lesion was provided in DICOM coordinates. Cancer was considered significant when the biopsy Gleason score was 7 or higher. The lesions were labelled with “TRUE” and “FALSE” for presence of clinically significant cancer, with a distribution of 67 clinically significant lesions (TRUE) and 214 clinically non-significant lesions (FALSE).

### 2.2. Feature Extraction

Manual segmentations of the whole prostate gland and of each lesion were performed independently by two radiologists (M.L., 10 years of experience, and A.U., radiology resident) on T2W and DW maps separately. The lesion segmentation on DWI was performed on the high b-value image and the whole gland segmentation was performed on the low b-value image. An example segmentation can be found in Figure 1. For each sample, one radiologist’s volume of interest (VOI) was randomly chosen to be included in the final dataset. Radiomic features were extracted using the package Pyradiomics (version 3.0) [23] in Python (v. 3.7.9; https://www.python.org/ (accessed on 28 September 2020)). 14 shape features, 18 firstorder features and 75 texture features were extracted from the VOI of three MRI modalities, T2W, DWI and ADC, resulting in a total of 321 features extracted. In the feature extraction of the ADC map, the mask drawn on the DWI was used. The mathematical expressions and semantic meanings of the features extracted can be found at https://pyradiomics.readthedocs.io/en/latest/ (accessed on 28 September 2020).

### 2.3. Dataset Construction

The features extracted from a lesion mask VOI constituted the Lesion Dataset. The features extracted from a whole gland mask VOI constituted the Gland Dataset. A Gland was considered to have clinically significant PCa if at least one of its lesions is clinically significant. From the previous datasets, two additional datasets were constructed:Lesion Features with Anatomical Zone dataset—A dataset composed of lesion features plus features describing the anatomical location of the lesion.Single-Lesion Whole Gland Features dataset—A truncated dataset composed of patients from the Gland dataset that had one only lesion.

The description of the datasets can be found in the Table 1.

The train/test split was performed with the train_test_split() function of the Python scikitlearn package (version 0.23.2; https://scikitlearn.org/ (accessed on 28 September 2020)) [24,25,26]. The hold out test sets consisted of 25% randomly selected samples from the original datasets and the split was stratified so that both train and test sets have the same proportion of True labels. The lesion train sets constituted of 210 lesions (51 clinically significant lesions and 159 clinically non-significant lesions) and test sets constituted of 71 lesions (16 clinically significant lesions and 55 clinically non-significant lesions). The gland train set constituted of 137 glands (48 clinically significant glands and 89 clinically non-significant glands) and test set constituted of 46 glands (15 clinically significant glands and 31 clinically non-significant glands). The single-lesion whole gland features dataset was not split into train and test set, due to its already reduced number of samples, and was only validated internally.

### 2.4. Feature Stability to Segmentation

Features that are highly dependent on segmentation margins, will not be stable predictors, since they easily change depending on the radiologist that performed the segmentation. Features extracted from the VOIs created by both radiologists were compared with Intraclass correlation coefficient (ICC). The ICC used was a two-way, single rater, absolute agreement ICC model (ICC 2.1) [27]. Features with ICC 95% confidence interval lower limit over 0.8 were considered to be robust to segmentation and were kept for further analysis. This analysis was performed in Python (v. 3.7.9; https://www.python.org/(accessed on 28 September 2020)) with the package icc (https://pypi.org/project/icc/ (accessed on 28 September 2020)).

### 2.5. Zero or Near-Zero Variance

Zero and nearzero variance analysis was performed with the nearZeroVar() function of the R caret package (version 6.086; https://topepo.github.io/caret/ (accessed on 28 September 2020)) [28].

### 2.6. Outlier Detection

In order to identify outliers, the local outlier factor (LOF) was used. Since scale affects the distance function, the data was normalized before applying the LOF algorithm. Samples with LOF over 2 were removed from the original not normalized dataset. Outlier detection and removal was performed inside cross validation with the software RapidMiner Studio (version 9.9; https://rapidminer.com/ (accessed on 28 September 2020)) [29].

### 2.7. Feature Correlation

The feature correlation analysis was performed inside cross validation on RapidMiner Studio (version 9.9; https://rapidminer.com/ (accessed on 28 September 2020)) [29] with the operator “Remove Correlated Attributes”. This operator uses the Pearson correlation coefficient to compute the correlation between each pair of features. If a pair of features is found to have a correlation higher than the threshold, one of the features is randomly eliminated. The correlation threshold was a hyperparameter optimized during model training.

### 2.8. Feature Selection

Four feature selection algorithms were applied separately, and their performance compared. These algorithms were recursive feature elimination with support vector machine weighing (SVM-RFE), Boruta algorithm [30], minimum redundancy maximum relevance algorithm (mRMR) and LASSO regularization.

### 2.9. Model Development

In this work, different aspects of model development were assessed and compared (Figure 2). The machine learning algorithms were chosen so as to cover a wide range of machine learning algorithm types [31]. In total, 288 pipelines were produced, corresponding to the different combinations. Each was trained and validated according to the diagram in Figure 3.

Hyperparameter tuning was done in a nested cross-validation fashion with an exhaustive grid search. This was performed on RapidMiner Studio (version 9.9; https://rapidminer.com/ (accessed on 28 September 2020)) with the operator “Optimize Parameters (Grid)” [29]. In this work, we have chosen to optimize the F2-score, so as to take into account the higher cost of a false negative misclassification when compared to a false positive. Additionally, we report Cohen’s Kappa.

A subset of best classifiers was selected according to the 4-fold cross-validation F2 and Kappa performances, following the rule: $CV_{F2}>0.8\cap CV_{Kappa}>0.5$. These were applied to the holdout test set for validation.

### 2.10. Metric Volatility Analysis

The Gland, Lesion, and Lesion with anatomical location Datasets were each randomly split in training and testing sets in 50 different ways, according to 50 different random seeds. Each of the highest-ranking classifiers was then trained on each of the 50 training sets and validated through both cross-validation and each of the 50 holdout testing sets. The distribution of cross-validation and test set performance results was recorded for further analysis (Figure 4). This analysis was based on the metric volatility analysis performed by the Probatus package (https://ing-bank.github.io/probatus/ (accessed on 28 September 2020)).

### 2.11. Distribution Comparison Tests

All performance distributions were tested for normality using the Shapiro-Wilk test [32] and the D’Agostino K2 test [33]. For each classifier, the distribution of cross-validation performances was compared to the distribution of test set performances, to assess whether they belonged to the same distribution. Two statistical tests were used: the paired *t*-test and the Kolmogorov-Smirnov test [34]. Both tests behave like common hypothesis tests and the hypothesis were as follows:

**Hypothesis** **1 (H1).**
*The distributions of cross-validation and test set performances are identical.*


**Hypothesis** **2 (H2).**
*The distributions of cross-validation and test set performances are different.*


The significance level, α, was chosen to be 0.05.

## 3. Results

### 3.1. Feature Stability to Segmentation

In the Lesion Dataset, 154 features were found to be unstable, out of the total 321 features. While, in the Gland Dataset, 64 features were found to be unstable, out of the total 321 features. Among the unstable features, the feature type that was considered the most unstable was texture features (72.73% of the unstable lesion features and 84.38% of the unstable gland features) and the MRI modalities that showed the lower stability were DWI (45.45% of the unstable lesion features) and ADC (82.81% of the unstable gland features).

### 3.2. Zero or Near-Zero Variance

In the Lesion Dataset, out of the total 169 stable features, 2 features were found to have near-zero variance: DWI_original_glszm_GrayLevelNonUniformity and ADC_original_glszm_GrayLevelNonUniformity. While, in the Gland Dataset, no features were excluded.

### 3.3. Classifier Development

In Figure 5a–d, we can see the cross-validation performance of the same 288 pipelines grouped by the different pipeline aspects assessed in this work.

Overall, the Boruta algorithm (Figure 5a) did not perform as well as expected. Despite having a high cross-validation F2, most kappa values were extremely low, especially for pipelines trained on whole gland features. Pipelines trained with data that underwent SVM-RFE achieved an average cross-validation F2 of 0.7226 and Kappa of 0.3781. While the feature sets that underwent mRMR achieved average performances of 0.7071 on F2 and 0.4095 on Kappa. Overall, at this stage, SVM-RFE and mRMR pipelines show a similar average performance. Pipelines trained with data that underwent Lasso feature selection achieved an average cross-validation F2 of 0.643 and Kappa of 0.347, not performing, on average, as high as SVM-RFE and mRMR.

In Figure 5b, we can see that the average cross-validation performance results were higher on the models trained with sampled data on both F2 and Kappa, with average F2 of 0.7541 and Kappa of 0.3659 on the models trained with downsampled data and F2 of 0.8094 and Kappa of 0.3666 on the models trained with SMOTE data. As expected, the pipelines trained with the original imbalanced dataset performed lower with an average F2 of 0.4779 and Kappa of 0.2626.

On average the Naïve Bayes classifier (Figure 5c) achieved an F2 of 0.6573 and a Kappa of 0.3016, the Logistic regression classifier achieved an F2 of 0.6569 and a Kappa of 0.3058, the Logistic regression classifier with Elastic Net regularization achieved an F2 of 0.6984 and a Kappa of 0.3002, the Adaboosted Decision Tree classifier achieved an F2 of 0.6784 and a Kappa of 0.2931, the Random Forest classifier achieved an F2 of 0.6725 and a Kappa of 0.3914 and, finally, the Extreme Gradient Boost classifier achieved an F2 of 0.7226 and a Kappa of 0.3885. While the Random Forest and Extreme Gradient Boost classifiers seem to have performed, on average, higher than the remaining machine learning algorithms, both an ANOVA and Kruskal-Wallis test revealed no statistically significant difference.

In Figure 5d, we can see that, on average, classifiers trained with whole Gland radiomic features achieved a cross-validation performance of 0.7426 on F2 and of 0.351 on Kappa. While classifiers trained with the Lesion Dataset achieved an average cross-validation F2 of 0.6344 and a Kappa of 0.2749. The classifiers trained with the Lesion features with anatomical zone dataset achieved an average cross-validation F2 of 0.6682 and a Kappa of 0.3687. Finally, the classifiers trained with the single-lesion whole gland features dataset achieved an average cross-validation F2 of 0.7508 and a Kappa of 0.3806. Overall, the pipelines trained with whole gland features performed, on average, higher than the ones trained on lesion features, both in terms of Kappa and of F2.

### 3.4. Best Classifiers Validation

Figure 6 shows the 26 models that satisfied the condition: $CV_{F2}>0.8\cap CV_{Kappa}>0.5$, as well as their performance on the cross-validation setting and hold-out test set. 65% of these are models trained on whole gland features. All of the best models were trained on data that underwent some kind of sampling: 42% downsampled data and 58% SMOTE data. Regarding feature selection, 31% of the pipelines included SVM-RFE, 50% included mRMR, 15% included Lasso and 4% included Boruta. As for the machine learning algorithm, the large majority of best models are tree-based algorithms (73%) and the remaining models are logistic regressions with or without elastic net regularization and one Naïve Bayes pipeline.

Table 2 shows the performance of the best models on the cross-validation setting and on the hold out test set in terms of F2, Kappa, ROC-AUC, and AUPRC. In addition, it shows the difference between cross-validation and test set performance, Δ. The models where this difference is closest to zero are the least overfitted models. There seems to be a cluster of overfitted models on the bottom of the table (in darker red). These correspond to the pipelines trained with Lesion data.

### 3.5. Metric Volatility Analysis

The mean and standard deviation values were calculated for each performance metric and each classifier, as well as the Δ values (Table 3). The latter are shown in Table 4, where each column is individually colour-coded from lowest value, in green, to highest value, in red. As previously, there seems to be a cluster of overfitted models on the bottom of the table (in darker red). These correspond to the pipelines trained with Lesion data. Three clusters of lower Δ can be found in green, these correspond to the pipelines where downsampling of the majority class was performed.

### 3.6. Distribution Comparison Tests

Out of 26 classifiers, 19 classifiers displayed a significant difference between the F2 test set performance distribution and the F2 cross-validation performance distribution, 5 classifiers displayed no significant difference between the test set performance distribution and the cross-validation performance distribution, 1 classifier displayed a significant difference on the Kolmogorov-Smirnov test but no difference on the paired *t*-test and 1 classifier displayed a significant difference on the Kolmogorov-Smirnov test but inconclusive results on the paired *t*-test.

Out of 26 classifiers, 15 classifiers displayed a significant difference between the Kappa test set performance distribution and the Kappa cross-validation performance distribution, 8 classifiers displayed no significant difference between the test set performance distribution and the cross-validation performance distribution, 1 classifier displayed a significant difference on the Kolmogorov-Smirnov test but no difference on the paired *t*-test, 1 classifier displayed a significant difference on the paired *t*-test test but no difference on the Kolmogorov-Smirnov and 1 classifier displayed a significant difference on the KolmogorovSmirnov test but inconclusive results on the paired *t*-test.

Five classifiers displayed no significant difference between the cross-validation performance and the test set performance on both performance metrics, these were: G_D_SVM-RFE_XGB, G_D_mRMR_XGB, G_D_Lasso_DT, G_D_Lasso_RF, and G_D_Lasso_XGB. These were also among the classifiers found to be least overfitted previously, supporting those results. The performance distributions of these 5 classifiers can be found in Figure 7.

## 4. Discussion

In this work, an extensive analysis of different dimensions of a machine learning pipeline were assessed and their performance compared.

We started by assessing the radiomic features’ stability to segmentation, where we found that the whole-gland features seem to be considerably more robust than lesion features (approximately 50% of lesion features were found to be unstable, compared to approximately 20% of gland features being unstable). This was expected since there is a lot more inter-reader variability in determining lesion borders when compared to whole gland borders.

Regarding feature selection, a low performance was unexpectedly observed from the pipelines that applied Boruta feature selection. These showed a high F2, because the model would classify the large majority of samples as the minority class, leading to a high recall. However, the low Kappa score makes it clear that these were not useful models. It was observed that the Boruta algorithm found very few features that were better predictors than the random versions of themselves. Hence, it is hypothesised that the number of features selected by the Boruta algorithm (around three features) was not enough to build a meaningful radiomics signature, which led to the poor results.

The pipelines where sampling was applied performed higher than the ones where no sampling was done, whether it was downsampling of the majority class or upsampling of the minority class with SMOTE. This was expected since training a model with balanced data gives it equal opportunities to learn from both classes. While the pipelines where SMOTE upsampling was performed seem to slightly outperform downsampling of the majority class, the volatility analysis showed that the latter produces classifiers that are consistently less overfitted and more reliable. This can be explained by the fact that SMOTE generates synthetic samples from the existing samples in the dataset. Thus, we are forcing the model to learn more from the same data, increasing the model’s confidence in random variability, or noise, present in the data, which results in the overfitted behaviour.

Among the most interesting findings is the higher performance of models trained with radiomic features extracted from the whole gland VOI, as well as their higher reliability and lower overfitting. This suggests that the areas surrounding tumorous lesions might offer relevant information regarding their overall aggressiveness in the form of Gleason score. In addition to suggesting that the monotonous lesion segmentation work performed by radiologists may not be necessary or even be harming to the radiomics signature. However, it is of note that a few patients had more than one lesion. If these multiple lesions have the same clinical significance (same target label), then it seems reasonable that the model performs higher with gland features since it has more information pointing to the correct label. In order to make a fair comparison between the performance of both types of input data, the single-lesion whole gland dataset was created, including only patients with a single lesion. The performance results obtained with this smaller dataset confirm the suspicions above, that whole gland features produce more reliable machine learning models than lesion features. Additionally, the volatility analysis showed that the Lesion-based models seem to be the most overfitted, which supports the previews findings. This is not the first paper of its kind to report the performance of whole-gland radiomic features to predict PCa clinical significance [35,36].

As a final note, it is important to point out that given so many pipeline combinations we have to assume that it is possible to find one that performs well by chance. Statistically speaking, we could remedy this by doing something similar to a multiple comparisons p-value correction. However, at this point, we are not aware of such a correction for machine learning performance metrics.

Regarding the MR sequences used, DW images and ADC maps, although related, do not offer the same information. DW images quantify the diffusion of water molecules in the tissue. Their sensitivity to diffusion is regulated by the b-value, with a high b-value allowing the distinction between healthy tissue and tumorous tissue (where there is a much higher restriction to diffusion). While ADC maps show the rate of variation of the DWI signal intensity with respect to a change in b-values. Here, it is known that PCa signal intensity on DWI decreases slower than healthy tissue’s signal intensity. Thus, the rate of variation will be lower and PCa will appear hypointense on the ADC map. The information provided by DWI and ADC is different and, so, not redundant, which is why we include both in our analysis.

In terms of literature comparison, the dataset used in this study has been widely used, both within and out of the SPIE-AAPM-NCI PROSTATEx challenge. In this setting, the highest performing classifier achieved an AUC of 0.87 [37]. Even though, we felt that the AUC was not the most appropriate metric to optimize, we still achieved AUCs of up to 0.90 while optimizing the F2-score. Recent literature has shown the growing interest in PSMA PET radiomics for the classification of PCa’s clinical significance [38,39], so it would be interesting to assess in the future how PET radiomics would perform in the context of this study.

This study has some limitations. First, this was a retrospective study and, so, a multicentre prospective analysis should be carried out to validate these results and investigate the impact these predictive models have on patient outcome. Second, only T2W, DWI, and ADC sequences were used. Other sequences, such as dynamic contrast enhanced MRI, could be worth exploring. Third, only one set of MRI sequences was evaluated per patient, so we were unable to evaluate the temporal stability of the radiomic features. Fourth, although the overall class imbalance was addressed through downsampling of the majority class or SMOTE upsampling of the minority class, we did not address the imbalanced nature of the anatomical location of lesions, with the large majority of lesions belonging to the PZ. It would be interesting to investigate the model’s performance on the different anatomical zones independently. Fifth, the use of a publicly available dataset increased transparency but limited our access to clinical data, such as PSA levels, patient age, or PI-RADS score, which are a fundamental component of a clinician’s assessment, but could not be included in our model. Sixth, dataset quality issues were not addressed, such as the sometimes imperfect location of the centroid of each lesion [40]. Seventh, despite the effort of performing a metric volatility analysis, proper assessment of real-world clinical performance is only possible through external validation. This important validation step will be addressed in future work. Finally, inherent to the Gleason system is the subjectivity of cancer grading, so we must keep in mind that the gold standard used in this study is subject to human error and inter or intra-observer variability. In addition to this, the definition of clinical significance might be based on more than Gleason score alone, and variables such as tumour volume or tumour category might be of relevance.

## 5. Conclusions

In conclusion, our results further confirm the validity of MRI-based radiomic features in the identification of clinically significant prostate cancer. Additionally, we highlight the higher performance of models trained with whole gland radiomic features, as well as their higher stability and lower overfitting, when compared to lesion VOI radiomic features.

## Figures and Tables

**Figure 1 cancers-13-06065-f001:**
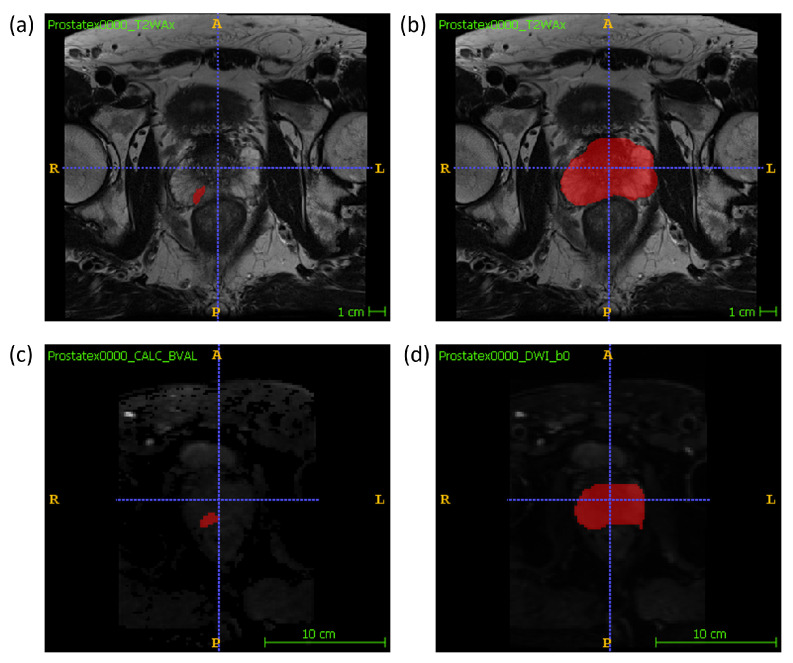
An example of the manual segmentation of lesions and glands performed in this study on T2W and DW sequences. (**a**) lesion segmentation on T2W; (**b**) gland segmentation on T2w; (**c**) lesion segmentation on high b-value DWI; (**d**) gland segmentation on b-value = 0 DWI.

**Figure 2 cancers-13-06065-f002:**
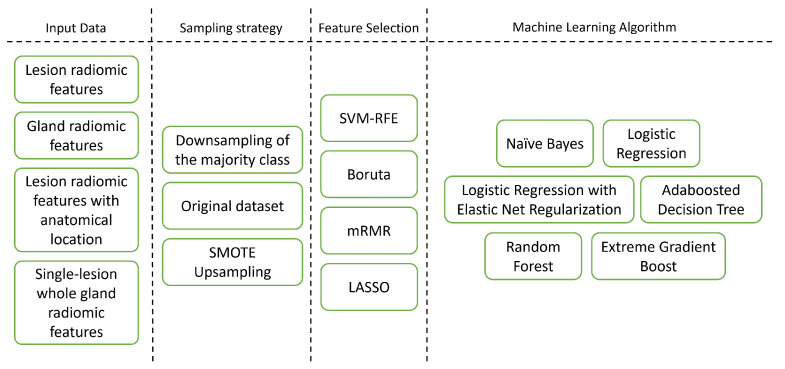
Different pipeline dimensions explored in this study.

**Figure 3 cancers-13-06065-f003:**
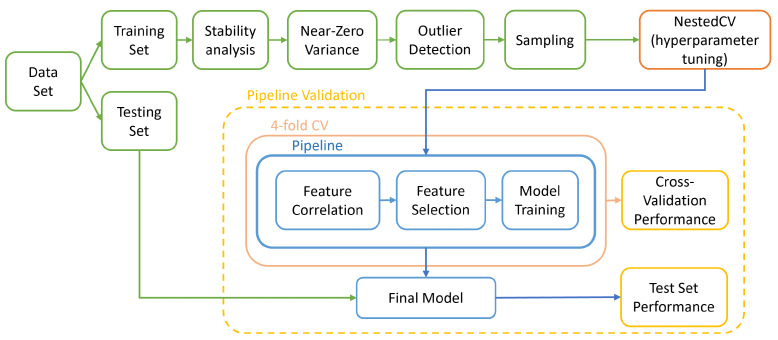
Overall pipeline followed in this study to train and validate models.

**Figure 4 cancers-13-06065-f004:**
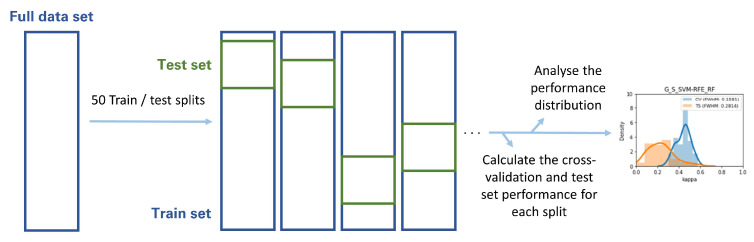
Methodology followed in the metric volatility analysis.

**Figure 5 cancers-13-06065-f005:**
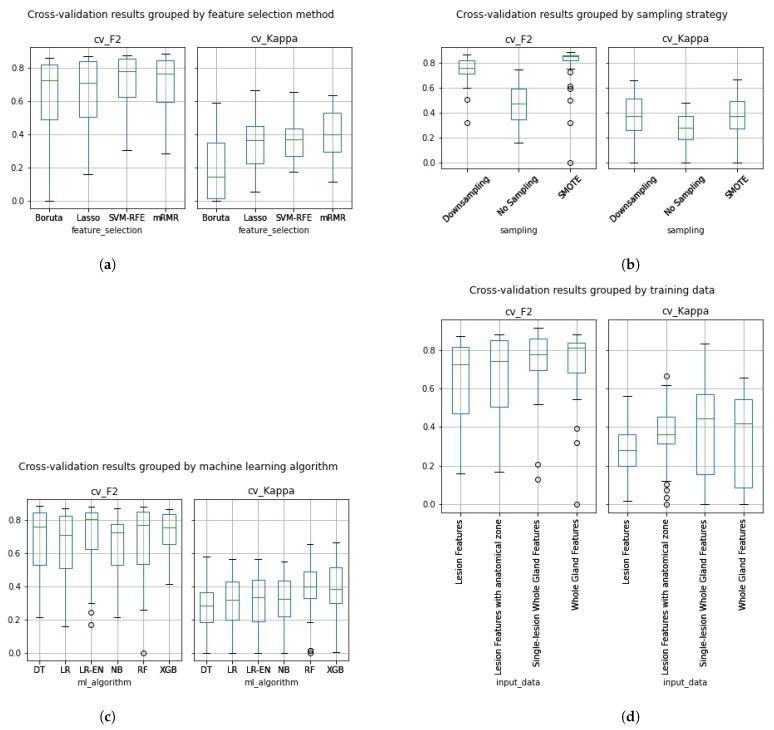
Cross-validation F2 and Kappa performance results grouped by (**a**) feature selection method, (**b**) sampling strategy, (**c**) machine learning algorithm and (**d**) type of input data.

**Figure 6 cancers-13-06065-f006:**
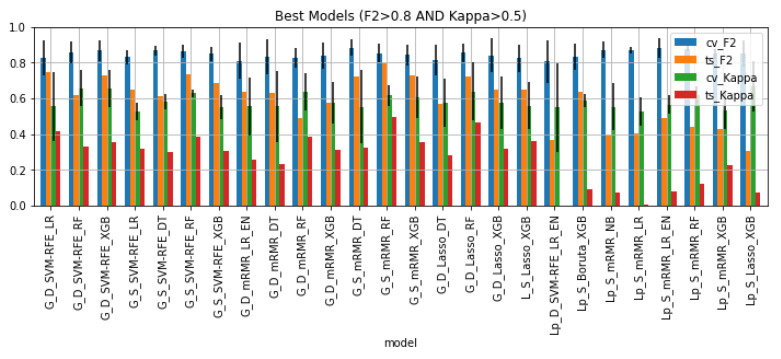
Performance of the best classifiers on the cross-validation setting and hold out test set in terms of F2 and Kappa.

**Figure 7 cancers-13-06065-f007:**
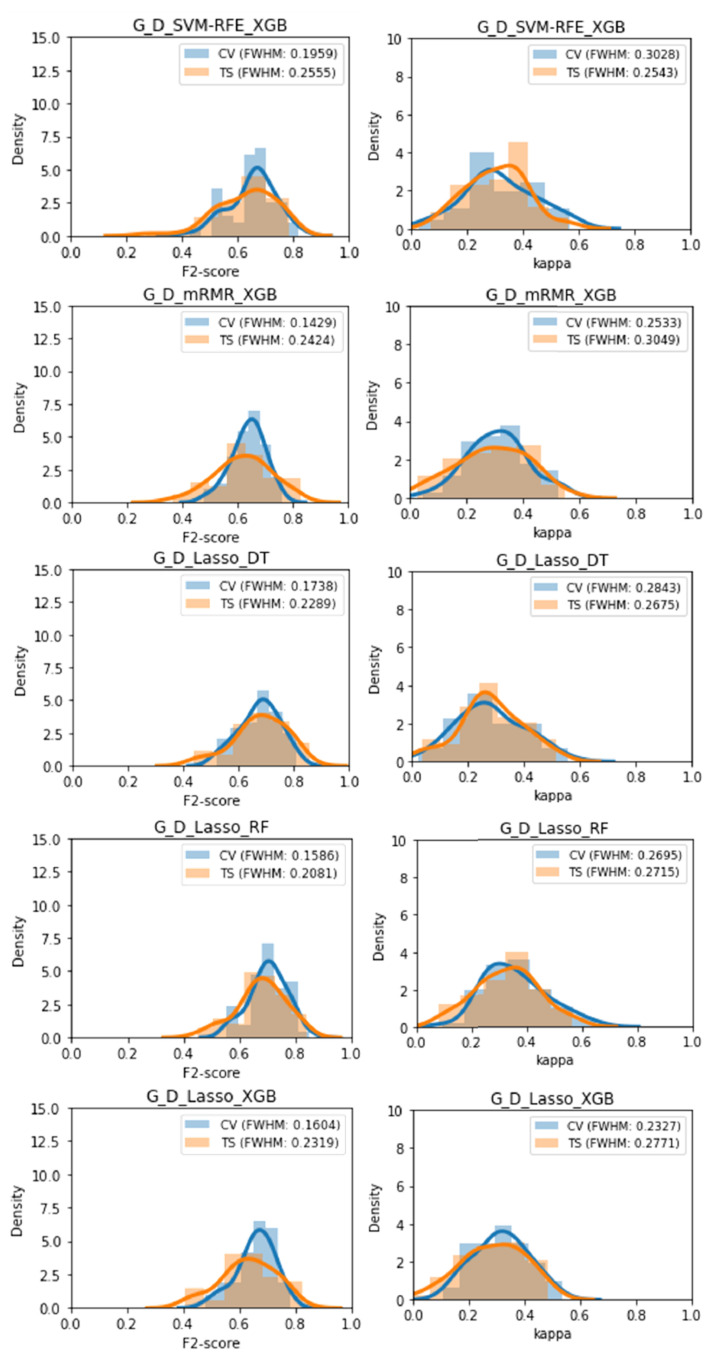
Distribution of F2 and Kappa performances obtained during the volatility analysis for each of the 5 classifiers with no statistically significant overfitting.

**Table 1 cancers-13-06065-t001:** Size and label distribution of the datasets utilized in this study.

Dataset	Number of Features	Number of Clinically Significant Cases	Number of Clinically Non-Significant Cases	Total
Lesion Dataset	321	67	214	281
Lesion Features with Anatomical Zone Dataset	325	67	214	281
Gland Dataset	321	63	120	183
Single-Lesion Whole Gland Features Dataset	321	33	74	107

**Table 2 cancers-13-06065-t002:** Best classifiers’ cross-validation and test set performances, as well as the difference between cross-validation and test set performance, Δ. The performance columns are colour coded from highest value in green, to lowest value in white. The Δ columns are colour coded from lowest value in green to highest value in red.

Model	cv_F2	ts_F2	cv_Kappa	ts_Kappa	cv_AUC	ts_AUC	cv_AUPRC	ts_AUPRC	Δ F2	Δ Kappa	Δ AUC	Δ AUPRC
G_D_SVM-RFE_LR	0.826	**0.745**	0.555	**0.416**	0.765	0.772	0.68	0.518	0.081	0.139	−0.007	0.162
G_D_SVM-RFE_RF	0.859	0.618	0.657	0.333	0.857	0.843	0.787	0.734	0.241	0.324	0.014	0.053
**G_D_SVM-RFE_XGB**	0.868	**0.729**	0.655	0.354	0.859	0.753	0.792	0.545	0.139	0.301	0.106	0.247
G_S_SVM-RFE_LR	0.831	0.652	0.528	0.32	0.798	0.766	0.742	0.655	0.179	0.208	0.032	0.087
G_S_SVM-RFE_DT	0.868	0.611	0.584	0.301	0.806	0.746	0.545	0.449	0.257	0.283	0.06	0.096
**G_S_SVM-RFE_RF**	0.862	**0.737**	0.629	0.385	0.873	0.788	0.841	0.576	0.125	0.244	0.085	0.265
G_S_SVM-RFE_XGB	0.849	0.684	0.551	0.308	0.847	0.728	0.805	0.504	0.165	0.243	0.119	0.301
G_D_mRMR_LR_EN	0.812	0.638	0.557	0.26	0.789	0.755	0.724	0.53	0.174	0.297	0.034	0.194
G_D_mRMR_DT	0.836	0.632	0.556	0.231	0.767	0.634	0.636	0.404	0.204	0.325	0.133	0.232
G_D_mRMR_RF	0.827	0.488	0.636	0.385	0.789	0.757	0.683	0.737	0.339	0.251	0.032	−0.054
G_D_mRMR_XGB	0.84	0.575	0.576	0.314	0.808	0.719	0.718	0.485	0.265	0.262	0.089	0.233
G_S_mRMR_DT	0.884	**0.722**	0.554	0.325	0.778	0.691	0.405	0.271	0.162	0.229	0.087	0.134
**G_S_mRMR_RF**	0.853	**0.798**	0.618	**0.494**	0.841	0.847	0.8	0.642	0.055	0.124	−0.006	0.158
G_S_mRMR_XGB	0.844	**0.729**	0.607	0.354	0.814	0.783	0.766	0.576	0.115	0.253	0.031	0.19
G_D_Lasso_DT	0.815	0.568	0.574	0.282	0.808	0.71	0.696	0.346	0.247	0.292	0.098	0.35
**G_D_Lasso_RF**	0.855	**0.722**	0.638	**0.466**	0.826	0.824	0.754	0.659	0.133	0.172	0.002	0.095
G_D_Lasso_XGB	0.84	0.652	0.576	0.32	0.856	0.7	0.798	0.447	0.188	0.256	0.156	0.351
L_S_Lasso_XGB	0.826	0.652	0.56	0.363	0.855	0.755	0.844	0.54	0.174	0.197	0.1	0.304
Lp_D_SVM-RFE_LR_EN	0.806	0.368	0.55	0.001	0.786	0.581	0.706	0.812	0.438	0.549	0.205	−0.106
Lp_S_Boruta_XGB	0.833	0.64	0.591	0.091	0.874	0.646	0.861	0.874	0.193	0.5	0.228	−0.013
Lp_S_mRMR_NB	0.873	0.389	0.554	0.075	0.836	0.55	0.793	0.713	0.484	0.479	0.286	0.08
Lp_S_mRMR_LR	0.872	0.404	0.528	0.006	0.853	0.53	0.804	0.783	0.468	0.522	0.323	0.021
Lp_S_mRMR_LR_EN	0.882	0.49	0.566	0.078	0.849	0.667	0.783	0.862	0.392	0.488	0.182	−0.079
Lp_S_mRMR_RF	0.879	0.44	0.617	0.124	0.881	0.58	0.868	0.805	0.439	0.493	0.301	0.063
Lp_S_mRMR_XGB	0.85	0.427	0.534	0.227	0.864	0.697	0.845	0.871	0.423	0.307	0.167	−0.026
Lp_S_Lasso_XGB	0.852	0.305	0.667	0.073	0.904	0.634	0.907	0.846	0.547	0.594	0.27	0.061

**Table 3 cancers-13-06065-t003:** Mean and standard deviation values calculated for each performance metric and each classifier during the volatility analysis.

Models	F2				Kappa				AUC				AUPRC				Δ (CV - TS)			
	CV		TS		CV		TS		CV		TS		CV		TS					
	mean	std	mean	std	mean	std	mean	std	mean	std	mean	std	mean	std	mean	std	F2	Kappa	AUC	AUPRC
G_D_SVM-RFE_LR	0.6447	0.0582	0.6060	0.0910	0.3103	0.1095	0.2754	0.1177	0.7041	0.0607	0.7130	0.0646	0.6308	0.0465	0.6281	0.1745	0.0388	0.0349	−0.0089	0.0027
G_D_SVM-RFE_RF	0.6734	0.0651	0.6207	0.0957	0.3278	0.1174	0.2678	0.1199	0.7195	0.0782	0.7076	0.0688	0.6502	0.0584	0.6286	0.1638	0.0527	0.0601	0.0119	0.0215
G_D_SVM-RFE_XGB	0.6538	0.0832	0.6309	0.1085	0.3168	0.1286	0.3033	0.1080	0.7074	0.0722	0.7072	0.0640	0.6371	0.0565	0.6159	0.1838	0.0230	0.0135	0.0002	0.0212
G_S_SVM-RFE_LR	0.8011	0.0302	0.6944	0.0822	0.4376	0.0683	0.2762	0.1193	0.7721	0.0324	0.7102	0.0690	0.7156	0.0410	0.6245	0.1752	0.1067	0.1614	0.0619	0.0911
G_S_SVM-RFE_DT	0.7939	0.0312	0.6484	0.0990	0.3893	0.0827	0.1906	0.1280	0.7357	0.0487	0.6336	0.0824	0.5059	0.0912	0.4347	0.1954	0.1454	0.1987	0.1021	0.0712
G_S_SVM-RFE_RF	0.7967	0.0278	0.6318	0.0963	0.4422	0.0671	0.2311	0.1195	0.8122	0.0278	0.6838	0.0747	0.7662	0.0272	0.5959	0.1722	0.1649	0.2110	0.1284	0.1703
G_S_SVM-RFE_XGB	0.7509	0.0462	0.5627	0.1085	0.4599	0.0752	0.2364	0.1450	0.7986	0.0318	0.6718	0.0733	0.7469	0.0366	0.5786	0.1704	0.1881	0.2235	0.1268	0.1683
G_D_mRMR_LR_EN	0.6445	0.0546	0.6109	0.0731	0.3174	0.1084	0.2740	0.1015	0.7135	0.0699	0.7097	0.0624	0.6415	0.0585	0.6274	0.1777	0.0336	0.0434	0.0039	0.0141
G_D_mRMR_DT	0.6043	0.1085	0.6168	0.1652	0.2307	0.1250	0.2207	0.0960	0.6583	0.0834	0.6567	0.0606	0.5721	0.0812	0.5641	0.1742	−0.0125	0.0100	0.0016	0.0080
G_D_mRMR_RF	0.6985	0.0635	0.6594	0.0778	0.3715	0.1103	0.3279	0.1120	0.7330	0.0621	0.7360	0.0648	0.6578	0.0527	0.6570	0.1593	0.0390	0.0436	−0.0030	0.0007
G_D_mRMR_XGB	0.6381	0.0607	0.6188	0.1029	0.3091	0.1076	0.2907	0.1295	0.7058	0.0646	0.6976	0.0862	0.6343	0.0538	0.6175	0.1671	0.0193	0.0184	0.0082	0.0169
G_S_mRMR_DT	0.8257	0.0309	0.6744	0.0757	0.4070	0.0907	0.2260	0.0954	0.7191	0.0434	0.6411	0.0585	0.4480	0.0857	0.3913	0.1666	0.1513	0.1809	0.0780	0.0567
G_S_mRMR_RF	0.8204	0.0296	0.6669	0.0782	0.4850	0.0617	0.2757	0.1116	0.8318	0.0298	0.7283	0.0595	0.7810	0.0305	0.6645	0.1473	0.1535	0.2093	0.1035	0.1165
G_S_mRMR_XGB	0.7490	0.0487	0.5749	0.0967	0.4706	0.0825	0.2607	0.1193	0.8041	0.0357	0.6764	0.0699	0.7544	0.0377	0.5802	0.1835	0.1741	0.2099	0.1276	0.1741
G_D_Lasso_DT	0.6755	0.0738	0.6785	0.0972	0.2788	0.1207	0.2810	0.1136	0.6756	0.0688	0.6784	0.0638	0.5255	0.0748	0.4893	0.1560	−0.0030	−0.0021	−0.0028	0.0362
G_D_Lasso_RF	0.7027	0.0673	0.6779	0.0884	0.3570	0.1144	0.3266	0.1153	0.7213	0.0725	0.7376	0.0728	0.6489	0.0617	0.6530	0.1565	0.0249	0.0304	−0.0163	−0.0041
G_D_Lasso_XGB	0.6590	0.0681	0.6317	0.0985	0.3173	0.0988	0.2875	0.1177	0.7117	0.0703	0.7060	0.0712	0.6386	0.0598	0.6218	0.1838	0.0273	0.0298	0.0058	0.0168
L_S_Lasso_XGB	0.7987	0.0242	0.4141	0.0960	0.5295	0.0457	0.1490	0.0990	0.8500	0.0190	0.6176	0.0610	0.8208	0.0229	0.4277	0.1933	0.3846	0.3805	0.2324	0.3931
Lp_D_SVM-RFE_LR_EN	0.6065	0.0759	0.5396	0.1112	0.2910	0.0984	0.2417	0.1041	0.6913	0.0562	0.7028	0.0715	0.6280	0.0542	0.4944	0.1902	0.0668	0.0493	−0.0115	0.1336
Lp_S_Boruta_XGB	0.7907	0.0277	0.2506	0.2442	0.5480	0.0520	0.0046	0.1402	0.8617	0.0205	0.4881	0.1042	0.8365	0.0230	0.2717	0.1541	0.5401	0.5434	0.3736	0.5648
Lp_S_mRMR_NB	0.7704	0.0453	0.5169	0.1168	0.4336	0.0780	0.2827	0.1331	0.7850	0.0309	0.7045	0.0638	0.7399	0.0313	0.4364	0.2120	0.2534	0.1510	0.0805	0.3035
Lp_S_mRMR_LR	0.8427	0.0178	0.5514	0.1006	0.3930	0.0662	0.2398	0.1102	0.7892	0.0264	0.6803	0.0676	0.7446	0.0374	0.4711	0.1940	0.2913	0.1532	0.1089	0.2735
Lp_S_mRMR_LR_EN	0.8397	0.0194	0.5388	0.1064	0.3781	0.0794	0.2328	0.1240	0.7814	0.0300	0.6840	0.0708	0.7360	0.0373	0.4729	0.1942	0.3009	0.1453	0.0973	0.2631
Lp_S_mRMR_RF	0.8454	0.0203	0.5705	0.0807	0.4940	0.0711	0.2463	0.0945	0.8556	0.0216	0.6925	0.0618	0.8274	0.0228	0.4832	0.1935	0.2749	0.2477	0.1631	0.3441
Lp_S_mRMR_XGB	0.8000	0.0319	0.5152	0.1058	0.5530	0.0528	0.1861	0.1203	0.8560	0.0236	0.6608	0.0732	0.8260	0.0248	0.4621	0.1957	0.2848	0.3669	0.1952	0.3639
Lp_S_Lasso_XGB	0.8012	0.0234	0.5124	0.0990	0.5588	0.0465	0.1614	0.0955	0.8642	0.0211	0.6505	0.0594	0.8364	0.0228	0.4513	0.1910	0.2888	0.3974	0.2137	0.3851

**Table 4 cancers-13-06065-t004:** Delta values calculated for each performance metric and each classifier during the volatility analysis. Each column is individually colour-coded from lowest value, in green, to highest value, in red.

Models	Δ (CV - TS)			
	**F2**	**Kappa**	**AUC**	**AUPRC**
G_D_SVM-RFE_LR	0.0388	0.0349	−0.0089	0.0027
G_D_SVM-RFE_RF	0.0527	0.0601	0.0119	0.0215
G_D_SVM-RFE_XGB	0.0230	0.0135	0.0002	0.0212
G_S_SVM-RFE_LR	0.1067	0.1614	0.0619	0.0911
G_S_SVM-RFE_DT	0.1454	0.1987	0.1021	0.0712
G_S_SVM-RFE_RF	0.1649	0.2110	0.1284	0.1703
G_S_SVM-RFE_XGB	0.1881	0.2235	0.1268	0.1683
G_D_mRMR_LR_EN	0.0336	0.0434	0.0039	0.0141
G_D_mRMR_DT	−0.0125	0.0100	0.0016	0.0080
G_D_mRMR_RF	0.0390	0.0436	−0.0030	0.0007
G_D_mRMR_XGB	0.0193	0.0184	0.0082	0.0169
G_S_mRMR_DT	0.1513	0.1809	0.0780	0.0567
G_S_mRMR_RF	0.1535	0.2093	0.1035	0.1165
G_S_mRMR_XGB	0.1741	0.2099	0.1276	0.1741
G_D_Lasso_DT	−0.0030	−0.0021	−0.0028	0.0362
G_D_Lasso_RF	0.0249	0.0304	−0.0163	−0.0041
G_D_Lasso_XGB	0.0273	0.0298	0.0058	0.0168
L_S_Lasso_XGB	0.3846	0.3805	0.2324	0.3931
Lp_D_SVM-RFE_LR_EN	0.0668	0.0493	−0.0115	0.1336
Lp_S_Boruta_XGB	0.5401	0.5434	0.3736	0.5648
Lp_S_mRMR_NB	0.2534	0.1510	0.0805	0.3035
Lp_S_mRMR_LR	0.2913	0.1532	0.1089	0.2735
Lp_S_mRMR_LR_EN	0.3009	0.1453	0.0973	0.2631
Lp_S_mRMR_RF	0.2749	0.2477	0.1631	0.3441
Lp_S_mRMR_XGB	0.2848	0.3669	0.1952	0.3639
Lp_S_Lasso_XGB	0.2888	0.3974	0.2137	0.3851

## Data Availability

Data will be available upon reasonable request.

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
