# Peer review of "Prediction of Prostate Cancer Disease Aggressiveness Using Bi-Parametric Mri Radiomics"

_cancers, 2021, doi:10.3390/cancers13236065_

Round 1

Reviewer 1 Report

Dear Authors,

I reviewed the manuscript entitled “PREDICTION OF PROSTATE CANCER DISEASE AGGRESSIVENESS USING BI-PARAMETRIC MRI RADIOMICS”, that is a well written and well designed investigation that focused on the use of ML algorithm in prostate cancer. However, I have the following comments:

  1. The classification process of applying ML algorithms based on the radiomics features is the general process. At least one or more newly proposed features should be included and analyzed for the publication with novelty. If there are new features proposed in this paper, please emphasize and describe the features.

  1. Please add information about the image acquisition parameters (for example dimension, spacing, and the original scale of the image intensities?). it would be interesting to add the method of normalization and scale of voxel values.

  1. Please specify the experience of the radiologists involved in the analysis.

  1. Please report the source of magnetic resonance exams and possible approval of the ethics committee.

  1. Please specify the gleason score.

Reviewer 2 Report

Dear Authors,

congratulations for this interesting manuscript that deals with a hot topic. Despite the many strengths of this paper (original approach, robust pipeline, good sample size etc.) I have some remarks:

1)  Spectroscopy is no longer consider as a mandatory sequence for mpMRI since PI-RADS v2 was released and I cannot foresee a possible change in a different direction since it adds to much time and complexity with to little diagnostic benefit. Therefore, spectroscopy does not deserve to be mentioned in the manuscript.

2) Why were DW images used for feature extraction along with ADC maps? Wouldn't the information be redundant? Were high b values images selected for this purpose? These points might be an interest topic for discussion.

3) It is stated that segmentation was performed separately for DW and T2 images. Was it not complicated to segment the whole gland on non-anatomical images? I think the readers would find it easier to get insights of the segmentation process if it was visualized in a figure and further details were added.

4) The choice of using a publicly available dataset is commendable. However, public datasets are not perfect and it has been found that the specific dataset used in this study has some issues (e.g. location of the centroid of each lesion is not always perfect) [10.1016/j.ejrad.2021.109647]. I believe that the Authors should discuss this issue. Did they perform a quality check? Furthermore, previously published lesion and gland annotations are for this dataset are freely available [10.1016/j.ejrad.2021.109647]. Did the Authors consider validating their results on independent segmentations?

5) The Authors should provide additional information for readers in the background, especially regarding advantages and current role of bpMRI [10.2214/AJR.20.23219] as well as current status of machine learning for clinically significant prostate cancer detection [10.1007/s00330-020-07027-w].

6) The definition of clinically significance for prostate cancer varies slightly and might be based on more than gleason score alone. The Authors should discuss the implications of different definitions for the application of their results.

Reviewer 3 Report

The topic is of interest and the methodology seems adequate. The main limitation is the single use of a public dataset (as the authors stated).  I have only minor points

1.) PSMA PET is growing of interest and the authors should also refer to some works on PSMA PET radiomics for GS prediction.

2.) The authors defined GS 7 as clinical significant. However, one may say that GS7a is rather not relevant and GS7b is relevant. Did the authors try to also discriminate between GS7a and GS7b since this discrimination would have consequences for therapeutic approaches (e.g. ADT during radiotherapy).

Round 2

Reviewer 2 Report

I wish to commend the Authors for their efforts in revising their manuscript. All issues raised in my previous report have been satisfactorily addressed and I have no further concerns. Kudos